# Classification of Glomerulonephritis with CNN and Self-Attention Networks in Individual Glomeruli in Nephropathology

Gloria Bueno[*]
*University of Castilla-La Mancha*, ES

Anibal Pedraza
*University of Castilla-La Mancha*, ES

Israel Mateos-Aparicio-Ruiz
*University of Castilla-La Mancha*, ES

Hien Van Nguyen
*University of Houston*, USA

Nicola Altini
*Politecnico di Bari*, IT

Huy Q. Vo
*University of Houston*, USA

Dejan Dobi
*Fac. of Med., Semmelweis University*, HU

Jean-Baptiste Gibier
*Lille University Hospital*, FR

Alessandro Del Gobbo
*Ospedale Maggiore Policlinico*, IT

Lucia González
*Hospital General Univ. Ciudad Real*, ES

Loreto Gesualdo
*University of Bari Aldo Moro*, IT

Francesco Pesce
*Fatebenefratelli IT - Gemelli Isola*, IT

Michele Rossini
*University of Bari Aldo Moro*, IT

Avi Rosenberg
*Johns Hopkins University*, USA

Jan Ulrich Becker
*University Hospital of Cologne*, GE

*Abstract*—Despite the renal biopsy being the gold standard for diagnosing glomerulonephritis, this practice remains inaccessible for many patients worldwide. Nephropathologists typically combine microscopy, immunohistology, transmission electron microscopy, clinical information, and genetic studies for diagnosis. However, variability in nephropathology evaluation has hindered its integration with emerging technologies and personalized medicine. This study proposes the use of deep learning to extract significant features to distinguish glomerulonephritis from PAS sections without other modalities. To test this hypothesis, various AI methods were tested for classifying 12 common glomerulonephritis diagnoses. Finally, a sequential classification was implemented, initially characterizing sclerosed and non-sclerosed glomeruli using Swin-Transformers, followed by classifying the non-sclerosed glomeruli into 12 types of glomerulonephritis using ConvNeXt. The first step achieved an average Balanced Accuracy of 97% and an AUC of 0.96. In the second step, a Balanced Accuracy considering up to the top3 of 79.5% and an avarage AUCs of 0.76 were achieved. This study establishes a baseline for this challenging classification task, demonstrating promising results even on single PAS glomerular crops.

*Index Terms*—Classification of Glomerulonephritis, Deep Learning in Nephropathology, Self-Attention Architectures, Digital Pathology.

## I. INTRODUCTION

THE digitalization of pathology departments has increased the use of whole slide images (WSIs), improving teleconsultation, education, and image archiving. Progress in machine learning (ML) and hardware has enabled the integration of artificial intelligence (AI), developing tools to assist pathologists in image analysis. Within digital pathology, nephropathology stands out as a subspecialty that deals with low-incidence diseases. Nephropathology has also been impacted by advancements in digital pathology and AI, as well as the work of multidisciplinary teams [1].

Glomerular diseases are conditions that directly affect the glomeruli. Patients with these types of diseases present multiple signs and symptoms (e.g., edema, proteinuria, and hematuria) that are present in several of them. These diseases are often characterized by certain common histopathological properties as well. However, even with diagnosis done this way, patients with the same disease show significant variability in their presentation, progression, and response to treatment [2]. Therefore, the classification of glomerular diseases has often been described as complex by experts [3].

Renal biopsy is the gold standard for diagnosing glomerulonephritis, using conventional microscopy techniques, immunohistology, and transmission electron microscopy. However, this practice remains inaccessible for diagnosing many patients worldwide. Nephropathologists combine these observations with clinical information and genetic studies to diagnose and provide prognostic and theranostic information. Despite efforts to standardize biopsy processing and histopathological descriptors, evaluation in nephropathology remains variable, hindering its integration with emerging technologies and personalized medicine.

The hypothesis of this work is that classification methods

The authors acknowledge funding support from the HANS project (Ref. PID2021-127567NB-I00), funded by the Spanish Ministry of Science, Innovation, and Universities, as well as by the European Union NextGenerationEU/PRTR. Additional support was provided by the MICROSCOM project (GA: 101128153), funded by the European Commission.
[*]Corresponding author: gloria.bueno@uclm.es

based on deep learning models are capable of extracting significant features that allow distinguishing glomerulonephritis from PAS sections, without the need of using other modalities. To test this hypothesis and analyze the capabilities of these methods, various AI methods have been implemented for the classification of 12 common diagnoses of glomerulonephritis. These glomerulonephritis and the acronym used in this work are shown in Table I.

One of the most common applications is to classify glomeruli into sclerosed or non-sclerosed. For this purpose, CNNs have been primarily used, as in [4], [5], and [6]. In this latter publication, segmentation and classification of tubular structures are also performed. Regarding other pathologies, Altini *et al.* [7] uses the Oxford classification [8]–[10] for Immunoglobulin A-associated Nephropathy (IgAN), with datasets containing glomeruli from the 12 types of glomerulonephritis used in this study. Other recent works apply AI techniques to locate glomeruli, identify lesions (global and segmental sclerosis, crescents), and quantify glomerular cells on PAS-stained slides with accuracy values around 91% [11]. In Liu's work [12], an AI-driven framework is developed for quantifying glomerular volume and nodular mesangial sclerosis in diabetic kidney disease, achieving 0.899 accuracy for glomerulus classification and an AUROC score of 0.917 for disease prediction. In the work of Zhuang *et al.* [13] the accuracy issues of ML models for diagnosing and predicting the prognosis of IgA nephropathy is addressed by conducting a systematic review of publications up to February 2024, from Embase, Pubmed, Cochrane Library, and Web of Science. Analyzing 47 studies with 51,935 patients, the pooled C-index was found to be 0.902 for 27 diagnostic models and 0.838 for 144 prognostic models. New ML models perform comparably to traditional ones. Future models should aim to improve sensitivity, especially in moderate-risk prediction, and focus on clinical application.

As far as the authors know, no further studies have been conducted to classify glomeruli from renal biopsies into the 12 classes considered here.

## II. MATERIALS

A total of 786 renal biopsies with glomerulonephritis (GN) from seven participating institutions were analyzed. Of these, 461 biopsies came from different patients. Each renal biopsy was diagnosed by expert nephropathologists with one of the 12 types of glomerulonephritis listed in I. The biopsies for all diagnostic classes were randomly selected without consideration of any histopathological descriptors or histological subtypes. Expert nephropathologists assigned and retrospectively confirmed the diagnostic classes prior to inclusion, using a standardized diagnostic work-up that included paraffin histology, immunostaining for immunoglobulin heavy and light chains, and transmission electron microscopy. This thorough process, applied to all cases, ensured a robust and reliable ground truth for the entire dataset.

The biopsies used in this work come from three different datasets: the $1^{st}$, consisting of 725 renal biopsies from 400

TABLE I: GN pathologies present in the dataset

| Name | Label |
|---|---|
| Anti-glomerular Basement Membrane antibody GN | AMBGN |
| Anti-Neutrophil Cytoplasmic Antibody-associated GN | ANCA |
| C3-GlomeruloNephritis | C3-GN |
| Cryoglobulinemic GN | CryoGN |
| Dense Deposit Disease | DDD |
| Fibrillary GN | Fibrilar |
| Infection-associated GN | IAGN |
| Immunoglobulin A-associated GN | IgAGN |
| Membranous nephropathy | Membranous |
| idiopathic MembranoProliferative GN | MPGN |
| Proliferative GN Monoclonal Immunoglobulin Deposits | PGNMID |
| Systemic Lupus Erythematodes-associated GN class IV | SLEGN-IV |

different patients, (dataset A); the $2^{nd}$ comprising 50 renal biopsies from different patients, where only the sclerotic glomeruli were used (dataset B); and the $3^{rd}$ (dataset C), used as a hold-out for testing, which consists of 61 renal biopsies from different patients. Datasets A and C contain the 12 types of GN.

The renal biopsies were digitized (WSIs) using various scanners, including the Hamamatsu NanoZoomer, the Axioscan (Carl Zeiss), the Pannoramic Midi Slide Scanner (3DHIS-TECH), and the Aperio CS2 (Leica). All of the WSIs use Periodic Acid-Schiff (PAS) staining. The glomeruli in the images were automatically detected using the segmentation model described in *[Ref. hidden for double-blind review]*, implemented by the co-authors, and reviewed by expert pathologists. For each glomerulus, a rectangular crop was extracted, leaving a margin of 300 pixels on each side. In total, 12,969 PAS-stained glomerular crops were obtained. Dataset A, used for training and validation, comprises 10,128 crops. Dataset B, used for training and validation of sclerotic glomeruli, comprises 1,170 crops. Dataset C, used for testing, comprises 1,671 crops. Each crop in datasets A and C retains its original diagnostic label corresponding to one of the 12 classes. Dataset A also contains sclerotic glomeruli (see Figure 1 for the class distribution of glomeruli crops across the entire datasets).

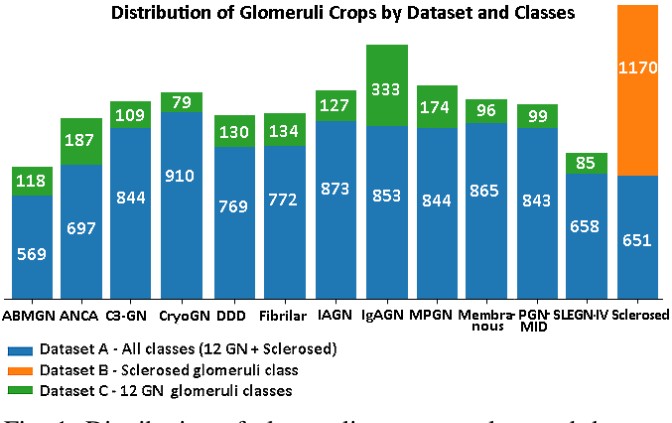

Fig. 1: Distribution of glomeruli crops per class and datasets

Figure 2 shows examples of glomeruli for each of the classes. It can be observed the variability in glomeruli, such as differences in texture and colour, even among glomeruli of the

same class. The variety of colors present in the images even when using the same stain, as well as the subtle differences at first glance between the different classes is also observed.

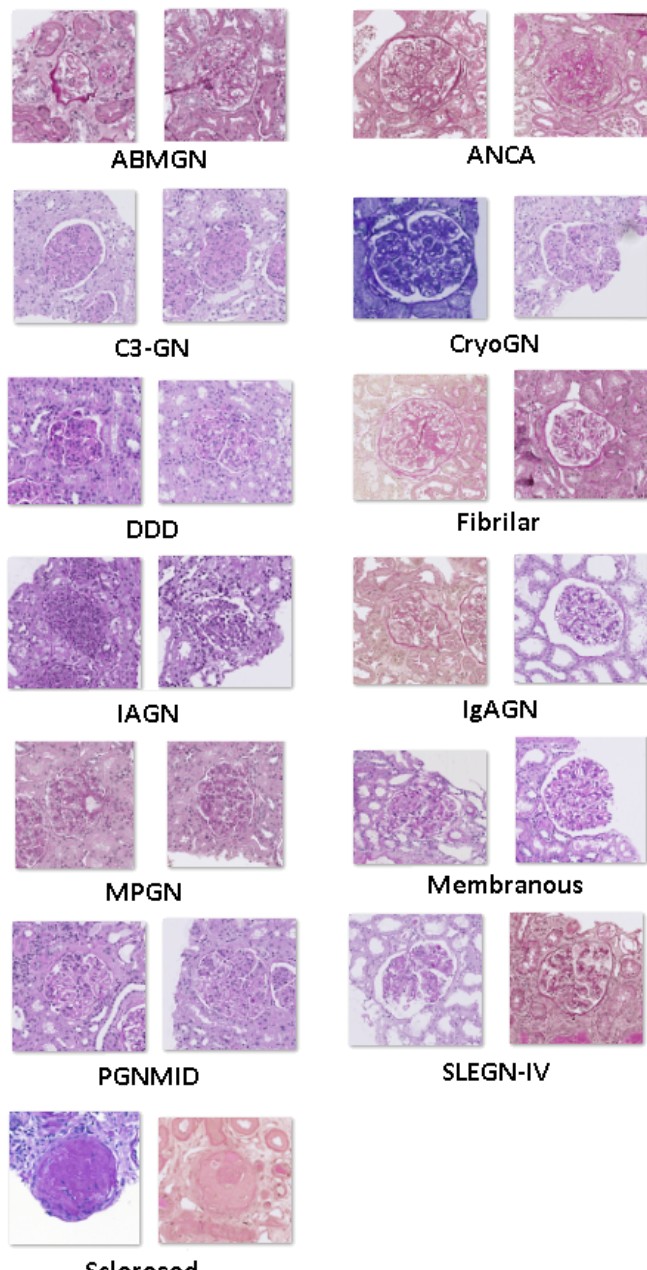

Fig. 2: Example images of each class of glomeruli. Two glomeruli from each class are shown

## III. METHODS

Several advanced classification networks were tested, these include: FocalNet, ConvNeXt, Swin-Transformers, and EfficientNet. These networks aim to optimize neural network architectures to improve efficiency and performance by using innovations in attention mechanisms and convolutions.

Due to the variability and the challenge of visually determining if individual glomeruli belong to the same slide, our objective was to incorporate more variability and broaden the dataset. Consequently, a partition was established at the glomerulus level, allocating 75% of the glomeruli crops for training, 15% for validation, and 10% for testing. The initial results from all networks were highly promising, with an average sensitivity of 0.98, specificity of 0.99 and a balanced accuracy of 0.99. However, when the dataset was partitioned at the slide level, to prevent information leakage across biopsies, the performance decreased drastically. It resulted in an average sensitivity of 0.2, an average specificity of 0.93, and a balanced accuracy of 0.63 with a standard deviation of 0.1 across all methods and classes.

Several tests were done with colour standardization and color transfer for data augmentation but the results only increased around 2%. Therefore, it was concluded that texture and the contextual relationships of the tissue plays a fundamental role in the diagnosis of these glomeruli crops.

Furthermore, activation maps were visualized using various techniques such as Grad-CAM++, ScoreCAM, FullGrad, LayerCAM, and XGradCAM to highlight areas of the images that significantly contribute to predicting specific classes. From the conducted tests, all methods show similar results as depicted in Figure 3.This figure shows two glomeruli, one of type DDD and one of type IAGN, along with activation maps from the six methods. No specific area was identified by the network as relevant or contributing to the prediction. Figure 4 shows activation maps using Grad-CAM++ for different glomeruli types: panel a) shows a Cryoglobulinemic-type glomerulus misclassified as ANCA, while panel b) shows a correctly classified DDD-type glomerulus.

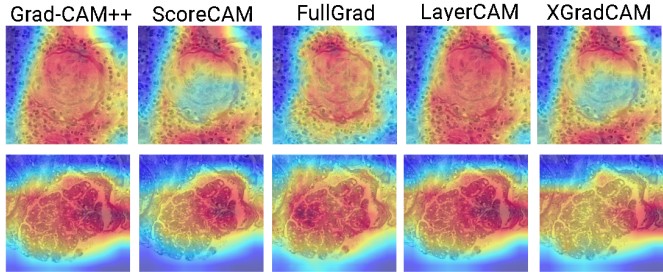

Fig. 3: Activation maps of glomeruli using Grad-CAM++, ScoreCAM, FullGrad, LayerCAM, and XGradCAM. $1^{st}$ row: DDD-type glomerulus; $2^{nd}$ row: IAGN-type glomerulus

By examining Figure 2, it can be observed that sclerosed glomeruli have unique textures distinct from those with GN. Therefore, a sequential process was designed and implemented: first categorizing glomeruli as sclerosed or non-sclerosed, then further dividing them into 12 distinct classes. The findings demonstrated a significant improvement, attaining a balanced accuracy of 0.97 in the initial classification of sclerosed and non-sclerosed glomeruli.

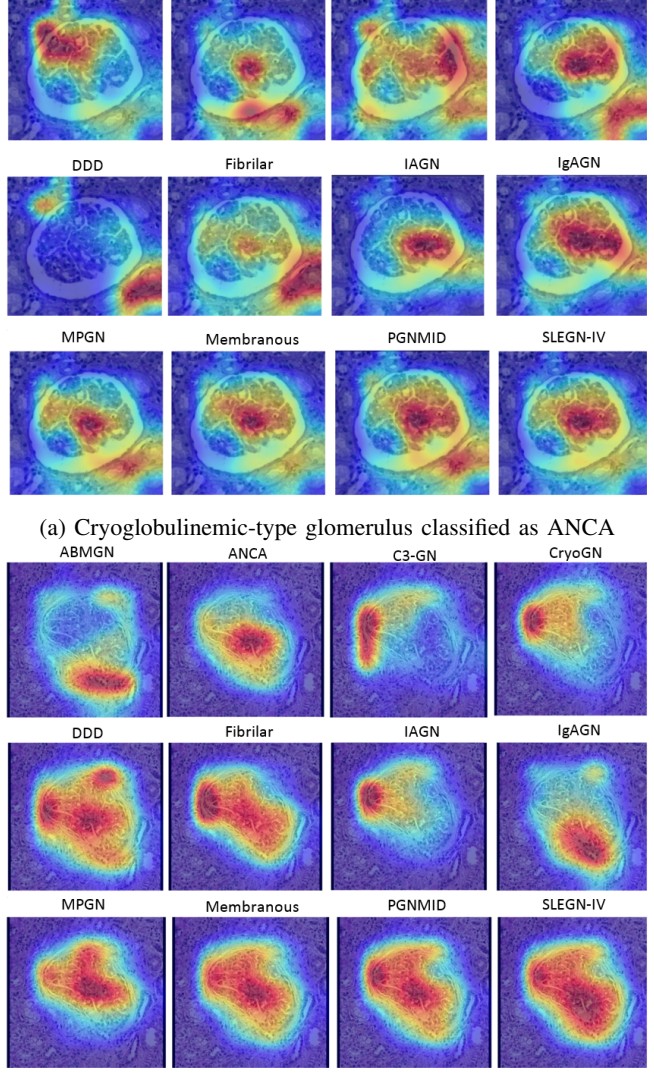

(a) Cryoglobulinemic-type glomerulus classified as ANCA

(b) DDD-type glomerulus classified correctly as DDD

Fig. 4: Activation maps using Grad-CAM++. Panel a) a Cryoglobulinemic-type glomerulus misclassified as ANCA and b) shows a correctly classified DDD-type glomerulus

### A. Datasets preparation

The preparation of datasets was as follows:

- Set of 12 Glomerulonephritis Pathologies

Sclerotic glomeruli were excluded, leaving a dataset of only non-sclerotic glomeruli crops from dataset A. This results in 12 classes for each GN pathology, without including glomerulosclerosis. The dataset partitions were created using a 5-fold cross-validation technique. However, the number of glomeruli in each set (training, validation, and test) varies across folds. This variation is due to a constraint applied when forming each fold. Specifically, glomeruli from the same WSI must be placed in the same set. This prevents models from learning to extract features shared by glomeruli from the same WSI. It ensures that models classify glomeruli based on pathology-related features, rather than features of the entire image. In addition, biopsies from the same patient are placed in the same set to prevent them from appearing in both the training and test sets.

The final distributions for the folds used in classifying the 12 GN pathologies were based on glomeruli crops from dataset A (see Figure 1), with allocations ranging from [71%, 14%, 15%] to [69%, 15%, 16%] for training, validation, and testing, respectively, across classes and folds.

- Sclerosed and Non-Sclerosed Set

The sclerosed and non-sclerosed glomeruli dataset was created based on the 12-glomerulonephritis dataset. The "Non-Sclerosed" class is a subset of glomeruli from the 12 pathologies, ensuring diverse representation. The partitions from each fold of the original dataset were maintained, adhering to WSI restrictions for consistency and allowing for independent model training. Initially, the test set showed an imbalance by including all glomeruli from the 12 glomerulonephritis pathologies. However, adding sclerosed glomeruli from dataset B balanced the classes. Since sclerosed and non-sclerosed glomeruli can share the same WSIs, WSI restrictions apply to both. This approach resulted in balanced datasets with a representative 'Non-Sclerosed' class. The distribution used for these two classed is shown in Table II for the best fold.

TABLE II: Distribution in training, validation, and test sets for the first fold of the sclerosed and non-sclerosed dataset

| Label | Train | Validation | Test | Total |
|-------|-------|------------|------|-------|
| Non-Sclerosed | 378 | 156 | 1670 | 2204 |
| Sclerosed | 376 | 156 | 1289 | 1821 |

### B. Base Models for Glomeruli Classification

Two base models for glomeruli classification were developed: one for classifying them into sclerosed and non-sclerosed, and another for classifying the non-sclerosed ones into 12 glomerulonephritis pathologies. The best-performing networks were EfficientNet [14], Swin Transformer [15], and ConvNeXt [16]. Below are the general characteristics and their training parameters. These parameters are the same for training the models for classification into sclerosed and non-sclerosed, as well as into the 12 pathologies. The only difference in the training is the dataset used.

- ConvNeXt

ConvNeXt is a convolutional network that incorporates improvements from transformers by adapting them into a purely convolutional solution. For example, to reduce dimensionality, it uses a convolution with a larger kernel size and a stride of 0, mimicking patch division; it also employs layer normalization instead of batch normalization. The architecture utilized in this project is ConvNeXt Base or ConvNeXt-B. The model is pretrained on ImageNet-1K and subsequently retrained using the glomeruli datasets with specific parameters: it employs the AdamW optimizer with an initial learning rate set to 0.001 and incorporates a weight decay of 0.05. Additional settings include an epsilon value of $10^{-8}$, $\beta_1$ of 0.9, and $\beta_2$ of 0.999.

The learning rate policy follows a cosine annealing approach, starting with a warmup phase of 20 epochs using a linear warmup type and a rate of 0.001. The minimum learning rate is established at 0.01, with a maximum training duration of 300 epochs and a batch size of 64. Input images for the model are resized to $224 \times 224$ pixels and normalized using ImageNet's mean and standard deviation. Training incorporates two data augmentation techniques: RandAugment [17] and RandomErasing [18].

- Swin-Transformers

Swin-Transformers use the sliding window mechanism which allows for better capture of local and global features, as well as improving the efficiency and scalability of transformers. Unlike the basic Vision Transformer (ViT), where the outputs of each layer maintain the same dimension, in Swin the patches become progressively smaller throughout the network (hierarchical structure). The version used in this work is Swin-T (Tiny), with the following parameters: Optimizer AdamW, Initial learning rate 0.001, Weight decay 0.05, $\epsilon$ $10^{-8}$, $\beta_1$ 0.9, $\beta_2$ 0.999, Learning rate policy: Cyclic with cosine annealing, Warm-up epochs 20, Warm-up type: Linear, Warm-up rate 0.001, Minimum learning rate 0.01, Maximum number of epochs 300, and Batch size 64. Additionally, the input images for the model are resized to $224 \times 224$ pixels and normalized using the mean and standard deviation of ImageNet. For training, two data augmentation policies are also used: RandAugment [17] and RandomErasing [18].

- EfficientNet

EfficientNet focuses on optimizing the size, depth, and number of channels of convolutional networks, providing an optimal balance between performance and computational efficiency. It uses a compound coefficient to scale the size of the network according to the size of the input. The model used in this work is EfficientNet-B, pretrained on ImageNet-1K and retrained using the glomeruli datasets with the following parameters: Optimizer SGD, Initial learning rate 0.1, Momentum 0.9, Weight decay 0.0001, Learning rate policy: Step decay; Steps (epochs) at 30, 60, and 90; Maximum number of epochs 300, and Batch size 4. Additionally, the input images for the model are resized to $224 \times 224$ pixels and normalized using the mean and standard deviation of ImageNet.

## IV. RESULTS

The classification of glomeruli into 12 classes showed improved results in the second stage after classifying and removing the sclerotic glomeruli. Swin-Transformers showed the best performance in the first stage while classifying sclerotic glomeruli versus non-sclerotic, and ConvNeXt showed the best performance in the second one.

### A. Base model for classification of glomeruli into sclerosed and non-sclerosed

The performance metrics for the base models classifying glomeruli into sclerosed and non-sclerosed are shown in Table III. These metrics correspond to the averages of the *test*

sets from each fold, in order to compare the overall performance of the models, regardless of the partition configuration.

Using *balanced accuracy* for comparison, Swin Transformer performs best with $0.9718 \pm 0.0144$, compared to $0.8604 \pm 0.0360$ for ConvNeXt and $0.7780 \pm 0.1636$ for EfficientNet. Similar results are observed when using *precision* as the reference metric. Regarding the ROC (Receiver Operating Characteristic) and their AUC (Area Under the Curve), Swin Transformer is again the best model, with an AUC of 1.00, followed by ConvNeXt with 0.99 and EfficientNet with 0.79. Figure 5 shows the two best models.

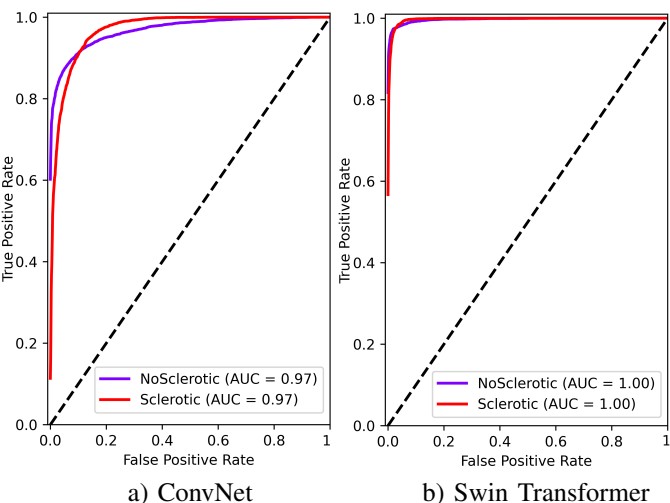

a) ConvNet      b) Swin Transformer

Fig. 5: ROC curves and average AUCs on the *test* set across 5 *folds* for the initial stage of glomerulus classification process, highlighting ConvNeXt and Swin Transformer as top performers in classifying Sclerosed and Non-sclerosed glomeruli.

### B. Base model for classifying non-sclerosed glomeruli into 12 renal glomerulonephritis pathologies

Figure 6 shows a visual comparison of the base models for classifying glomeruli into the 12 proposed pathologies. These metrics are the average of the 5 *folds*. Although only the model generated in one *fold* is finally used for testing with the hold-out dataset, this average is used to compare different methods, aiming for a comparison independent of the partitions taken. Taking *balanced accuracy* as the reference metric, the best-performing model is ConvNeXt, with an average for the 12 classes of $0.6418 \pm 0.0837$, compared to $0.6334 \pm 0.1080$ for Swin Transformer and $0.6009 \pm 0.0772$ for EfficientNet.

It was also observed that the model correctly identified an average balanced accuracy of 0.795 for all positive classes within the top 3 most likely. Thus, the results were analyzed considering that the model was correct if the true class was among the top-2 and top-3 predicted classes. The complete results for this model are illustrated in Table IV.

If we consider ROC curves and AUC, the difference between the models is higher. Once again, the top model is ConvNeXt, with an average AUC for the 12 classes of $0.7908 \pm 0.0831$, followed by Swin Transformer with $0.7625 \pm 0.0710$,

TABLE III: Results on the *test* set for the three baseline models for classifying glomeruli as sclerosed and non-sclerosed. The best result for the *sclerosed* class for each metric is highlighted in bold.

| Model | Class | Precision | Sensitivity | Specificity | F1-Score | Balanced Accuracy |
|---|---|---|---|---|---|---|
| ConvNeXt | Non-sclerosed | $0.8300 \pm 0.0372$ | $0.9542 \pm 0.0439$ | $0.7666 \pm 0.0706$ | $0.8870 \pm 0.0281$ | $0.8604 \pm 0.0360$ |
| | Sclerosed | $0.9386 \pm 0.0531$ | $0.7666 \pm 0.0706$ | $0.9542 \pm 0.0439$ | $0.8415 \pm 0.0443$ | $0.8604 \pm 0.0360$ |
| **Swin Transf.** | Non-sclerosed | $0.9702 \pm 0.0243$ | $0.9790 \pm 0.0055$ | $0.9647 \pm 0.0291$ | $0.9745 \pm 0.0122$ | $0.9718 \pm 0.0144$ |
| | **Sclerosed** | **$0.9752 \pm 0.0055$** | **$0.9647 \pm 0.0291$** | **$0.9790 \pm 0.0055$** | **$0.9697 \pm 0.0148$** | **$0.9718 \pm 0.0144$** |
| EfficientNet | Non-sclerosed | $0.7725 \pm 0.1594$ | $0.8881 \pm 0.1293$ | $0.6679 \pm 0.2426$ | $0.8219 \pm 0.1308$ | $0.7780 \pm 0.1636$ |
| | Sclerosed | $0.8376 \pm 0.2122$ | $0.6679 \pm 0.2426$ | $0.8881 \pm 0.1293$ | $0.7327 \pm 0.2089$ | $0.7780 \pm 0.1636$ |

TABLE IV: Complete results, from Top-1 to Top-3, on the test set of ConvNeXt for the classification of glomeruli into the 12 pathologies. The worst result is highlighted in red and the best in green.

| Class | Top-$k$ | Precision | Sensibility | Specificity | F1-Score | Balanced Accuracy |
|---|---|---|---|---|---|---|
| ABMGN | Top-1 | $0.3293 \pm 0.1296$ | $0.1756 \pm 0.0444$ | $0.965 \pm 0.0204$ | $0.2236 \pm 0.0611$ | $0.5703 \pm 0.0274$ |
| | Top-3 | $0.6971 \pm 0.1404$ | $0.5163 \pm 0.1797$ | $0.9792 \pm 0.0167$ | $0.5731 \pm 0.1547$ | $0.7477 \pm 0.0878$ |
| ANCA | Top-1 | $0.2117 \pm 0.1312$ | $0.2242 \pm 0.1364$ | $0.9372 \pm 0.0136$ | $0.2171 \pm 0.133$ | $0.5807 \pm 0.0733$ |
| | Top-3 | $0.5735 \pm 0.1353$ | $0.5499 \pm 0.1907$ | $0.9706 \pm 0.0075$ | $0.5589 \pm 0.1631$ | $0.7603 \pm 0.098$ |
| **C3-GN** | Top-1 | $0.1497 \pm 0.1343$ | $0.1246 \pm 0.1106$ | $0.9541 \pm 0.023$ | $0.1354 \pm 0.1203$ | $0.5393 \pm 0.0507$ |
| | Top-3 | $0.506 \pm 0.1702$ | $0.5105 \pm 0.1819$ | $0.9751 \pm 0.0138$ | $0.4072 \pm 0.2673$ | $0.6778 \pm 0.1371$ |
| CryoGN | Top-1 | $0.3745 \pm 0.2074$ | $0.3215 \pm 0.2428$ | $0.9429 \pm 0.0349$ | $0.3142 \pm 0.1629$ | $0.6322 \pm 0.1112$ |
| | Top-3 | $0.6258 \pm 0.1579$ | $0.5595 \pm 0.2504$ | $0.9632 \pm 0.0264$ | $0.5609 \pm 0.1471$ | $0.7613 \pm 0.1163$ |
| DDD | Top-1 | $0.5717 \pm 0.2014$ | $0.3561 \pm 0.2612$ | $0.9778 \pm 0.0139$ | $0.4095 \pm 0.2523$ | $0.6669 \pm 0.1288$ |
| | Top-3 | $0.8103 \pm 0.1369$ | $0.5808 \pm 0.2513$ | $0.986 \pm 0.0118$ | $0.6515 \pm 0.1896$ | $0.7834 \pm 0.1255$ |
| Fibrilar | Top-1 | $0.283 \pm 0.0837$ | $0.309 \pm 0.1714$ | $0.9383 \pm 0.0208$ | $0.2895 \pm 0.1231$ | $0.6236 \pm 0.0797$ |
| | Top-3 | $0.6007 \pm 0.0354$ | $0.5771 \pm 0.1034$ | $0.968 \pm 0.007$ | $0.5853 \pm 0.0602$ | $0.7726 \pm 0.0487$ |
| IAGN | Top-1 | $0.3622 \pm 0.2474$ | $0.3069 \pm 0.3608$ | $0.9589 \pm 0.0169$ | $0.3137 \pm 0.307$ | $0.6329 \pm 0.1762$ |
| | Top-3 | $0.6586 \pm 0.2653$ | $0.5285 \pm 0.3627$ | $0.9735 \pm 0.0169$ | $0.557 \pm 0.3078$ | $0.751 \pm 0.1808$ |
| **IgAGN** | Top-1 | $0.4383 \pm 0.0927$ | $0.6349 \pm 0.0676$ | $0.9201 \pm 0.0265$ | $0.5158 \pm 0.08$ | $0.7775 \pm 0.0397$ |
| | Top-3 | $0.6798 \pm 0.0779$ | $0.8795 \pm 0.0673$ | $0.96 \pm 0.0145$ | $0.7653 \pm 0.0674$ | $0.9197 \pm 0.0367$ |
| MPGN | Top-1 | $0.1742 \pm 0.0848$ | $0.2153 \pm 0.073$ | $0.8928 \pm 0.0384$ | $0.1876 \pm 0.0729$ | $0.5541 \pm 0.0446$ |
| | Top-3 | $0.4801 \pm 0.1589$ | $0.5912 \pm 0.1472$ | $0.9369 \pm 0.0252$ | $0.5281 \pm 0.1542$ | $0.7641 \pm 0.0834$ |
| **Membranous** | Top-1 | $0.6341 \pm 0.1504$ | $0.512 \pm 0.1902$ | $0.9683 \pm 0.0201$ | $0.5401 \pm 0.1166$ | $0.7402 \pm 0.0899$ |
| | Top-3 | $0.8326 \pm 0.0804$ | $0.8413 \pm 0.0982$ | $0.9834 \pm 0.0095$ | $0.8316 \pm 0.0479$ | $0.9123 \pm 0.0469$ |
| PGNMID | Top-1 | $0.3614 \pm 0.0482$ | $0.5227 \pm 0.1037$ | $0.9044 \pm 0.0239$ | $0.4219 \pm 0.0449$ | $0.7136 \pm 0.0452$ |
| | Top-3 | $0.5998 \pm 0.0458$ | $0.8418 \pm 0.104$ | $0.942 \pm 0.0138$ | $0.6959 \pm 0.027$ | $0.8919 \pm 0.0466$ |
| SLEGN-IV | Top-1 | $0.3324 \pm 0.1913$ | $0.4073 \pm 0.2555$ | $0.9325 \pm 0.024$ | $0.364 \pm 0.2163$ | $0.6699 \pm 0.1374$ |
| | Top-3 | $0.5334 \pm 0.2358$ | $0.6035 \pm 0.2465$ | $0.9568 \pm 0.0233$ | $0.5642 \pm 0.2389$ | $0.7801 \pm 0.1337$ |

and EfficientNet with $0.7317 \pm 0.0945$. Figure 7 displays the ROC curves and AUC of the top two methods.

Figure 7 shows superior results compared to Table IV because they assess different aspects with different methods. The ROC curve measures the model's ability to distinguish between positive and negative samples of a class at various classification thresholds. This indicates good ranking ability but not necessarily optimal classification performance at a fixed threshold. For all three models, performance varies widely across classes, especially when considering other metrics like precision. Taking ConvNeXt's precision as an example across each class, we observe results ranging from 0.63 for membranous nephropathy to 0.15 for C3-GN in top1. The low result for the latter can be justified because diagnosing this pathology typically requires other techniques such as immunofluorescence [19], [20].

However, results for top-3 improve for classification of all GN pathologies. The values increase for all metrics by between 0.21 and 0.35, with the standard deviation remaining consistent. For example, balanced accuracy in top-3 ranges from 0.68 to 0.92. The best-performing classes are Membranous, IgAGN, and PGNMID, where a balance between sensitivity and specificity is maintained, with sensitivity values around 0.84 and 0.87 and specificity ranging from 0.94 to 0.98.

After selecting ConvNeXt as the best model out of the three, the model generated from the best fold is evaluated with the hold-out dataset. This set is entirely independent from the others and consists of all images from database C. The results are shown in Figure 8. The overall performance is slightly lower, with a balanced accuracy of $0.577 \pm 0.074$, compared to $0.642 \pm 0.084$ previously. The ROC curves and their respective AUCs are shown in Figure 9. The average AUC values on the hold-out dataset are slightly lower, 0.69 compared to the previously obtained 0.79. However, classification improves slightly in 5 classes.

### C. Discussion

The results highlight the challenge of identifying key features among the 12 GN pathologies and the need for additional information such as immunofluorescence tests, for accurate diagnosis.. Manual classification with the hold-out dataset by four pathologists was performed to assess inter-pathologist variation and compare with ConvNeXt. The difficulty in classifying some pathologies was consistent, with higher agreement in top-3 results. Pathologists noted that some glomeruli crops could be classified into multiple classes visually. Therefore, certain pathologies require the use of additional sample pro-

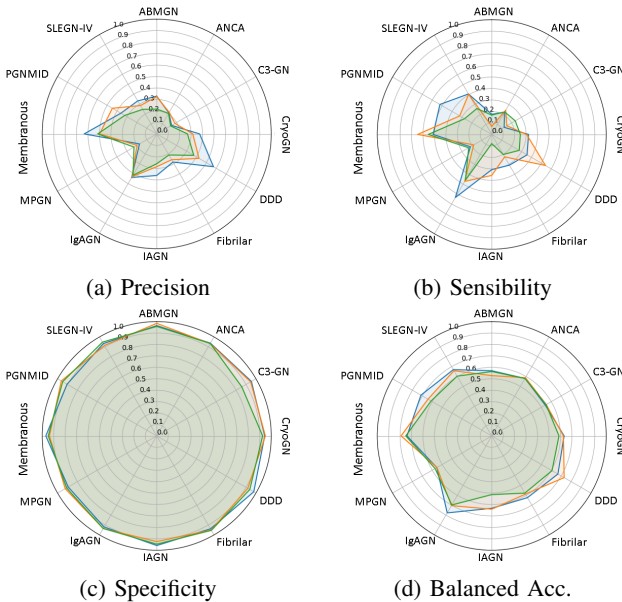

(a) Precision      (b) Sensibility

(c) Specificity      (d) Balanced Acc.

Fig. 6: Results on the *test* set for the three models classifying glomeruli into the 12 GN pathologies. In blue, ConvNeXt; in orange, Swin Transformer; and in green, EfficientNet.

cessing techniques, as both methods and experts are unable to distinguish between them using only PAS-stained images.

In all methods, performance varies greatly depending on the pathology. Some, such as C3-GN, consistently yield the worst results across all models, unlike Membranous, which always achieved the best results. Pathologies with poorer performance across all models often require additional tests for diagnosis or belong to subtypes within the same group of pathologies, such as MPGN, DDD, and C3-GN [19], [20]. This reaffirms not only the need to include other types of medical tests in training but also the necessity for a more differentiable taxonomy among pathologies at a medical level.

## V. CONCLUSIONS

Complete microscopic images were obtained and properly processed for use in deep learning models. Expert annotations were made on complete images, and glomerular crops were extracted and appropriately labeled. Different partitions were prepared using a custom variation of the 5-fold cross-validation technique, which were subsequently used to compare models by evaluating their performances independently of the partition configuration. Specific constraints were applied within the realm of histological images during dataset creation at the patient and WSI levels to prevent data leakage between sets. Additionally, the test sets were enriched with a completely different database, which also helped balance the dataset and test the sequential model implemented. Thus, a robust dataset was successfully created for the training and evaluation of deep learning models.

A sequential model was implemented to first classify glomeruli as sclerosed or non-sclerosed, then further classify non-sclerosed glomeruli into 12 GN pathologies. The models

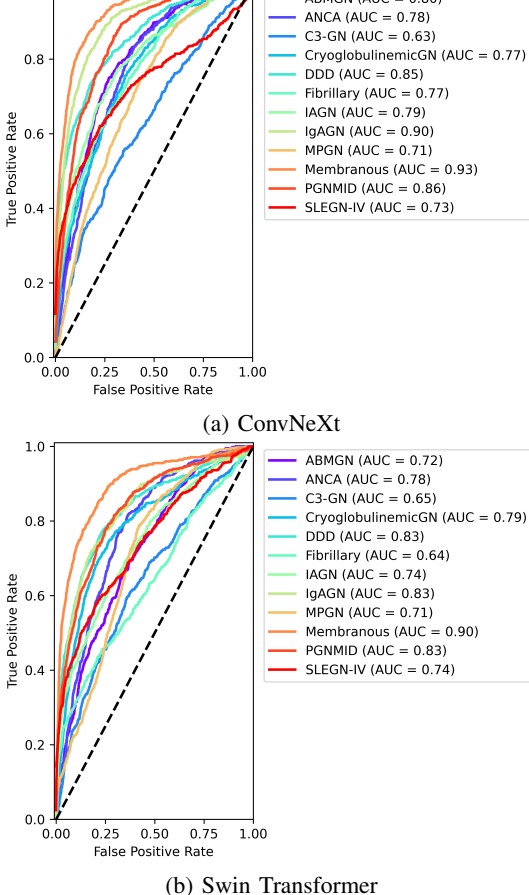

(a) ConvNeXt

(b) Swin Transformer

Fig. 7: ROC curves and average AUCs on the *test* set across 5 *folds* for the second stage of glomerulus classification process, highlighting ConvNeXt and Swin Transformer as top performers in the 12 glomerulonephritis classification.

were evaluated using average metrics across 5 folds, including precision, sensitivity, specificity, F1-score, balanced accuracy, and AUC from ROC analysis.

Swin Transformer was chosen as the best model for the first stage due to its superior overall performance, with a *balanced accuracy* of $0.9718 \pm 0.0144$ and an AUC of $1.00$. ConvNeXt was selected as the best model for the second stage to classify the 12 GN types, with an average balanced accuracy of $0.6418$ for top-1 and $0.795$ for top-3 across the test sets in the 5 folds, along with an AUC of $0.75$. In the hold-out dataset, this best model achieved an average balanced accuracy of $0.5774$ and an AUC of $0.6925$. The most accurately detected GN types were Membranous and IgAGN. The Membranous class achieved the best values in all metrics from top-1 to top-3 classification, with a balanced accuracy of $0.84$ and an average AUC of $0.86$ across all datasets.

It can be asserted that the model reliably diagnoses glomerulosclerosis and further distinguishes between 12 types of glomerulonephritis with consistent performance, particularly identifying the Membranous class.

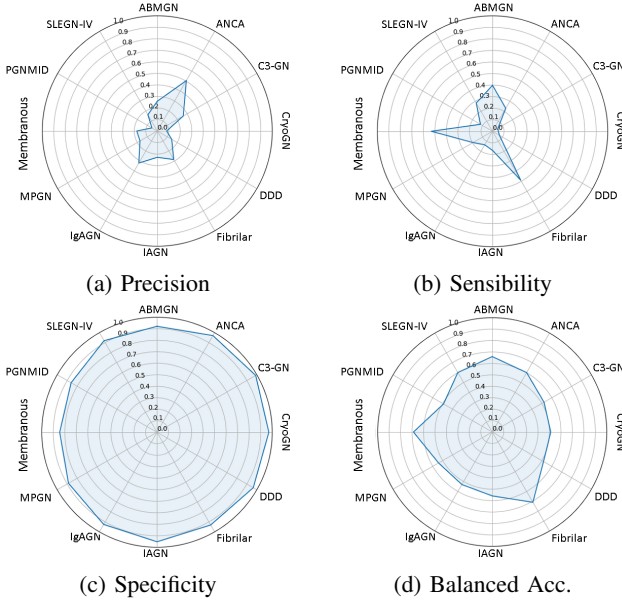

(a) Precision  (b) Sensibility

(c) Specificity  (d) Balanced Acc.

Fig. 8: Results on the hold-out dataset of the best base model for glomerulus classification into 12 GN pathologies. This model is the best fold of ConvNeXt.

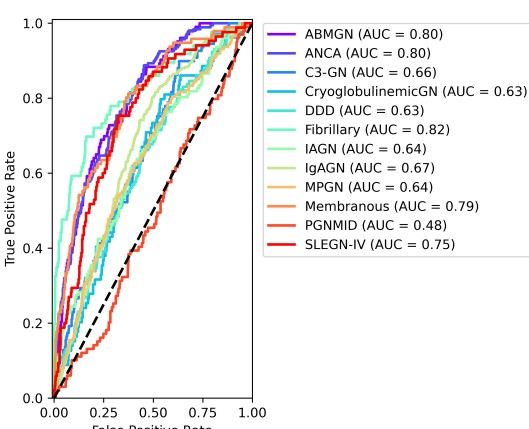

ABMGN (AUC = 0.80)
ANCA (AUC = 0.80)
C3-GN (AUC = 0.66)
CryoglobulinemicGN (AUC = 0.63)
DDD (AUC = 0.63)
Fibrillary (AUC = 0.82)
IAGN (AUC = 0.64)
IgAGN (AUC = 0.67)
MPGN (AUC = 0.64)
Membranous (AUC = 0.79)
PGNMID (AUC = 0.48)
SLEGN-IV (AUC = 0.75)

Fig. 9: ROC curves and AUCs on the hold-out dataset for the best base method of glomerulus classification across the 12 pathologies. The model is the top fold of ConvNeXt.

A QuPath extension has been developed with these models, aiming to ease its integration into clinical practice. The tool classifies glomeruli as sclerosed or non-sclerosed, and predict the most likely GN class from top-1 to top-3 for non-sclerosed cases. This proof-of-concept study establishes a baseline for this difficult classification task, which usually requires immunostains, electron microscopy and even clinical data.

### ACKNOWLEDGMENT

We would like to acknowledge the collaboration of Sándor Turkevi-Nagy from the Department of Pathology, Albert Szent-Györgyi Health Center, University of Szeged, Hungary; and Surya V. Seshan from Weill Cornell Medicine, New York, USA, for their assistance with validation and data provision.

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
