# OpenReview forum: "Classification of Glomerulonephritis with CNN and Self-Attention Networks in Individual Glomeruli in Nephropathology"
_IEEE.org/EMBS/BHI/2024/Conference — IEEE BHI'24_

### Official Review · Reviewer_nYcc · 2024-08-10
**review of "Classification of Glomerulonephritis with CNN and Self-Attention Networks in Individual Glomeruli in Nephropathology"**

**Overall Rating:** 5
**Confidence:** 4

**Other Quality Metrics:**

Clarity of writing: Good.
Clinical Significance: Good.
Methodological Novelty: Fair.
Experiments and Results: Fair.

**Questions For The Authors:**

The reason why choose 350 50, 61 when divided data into 3 dataset is not well explained.

How do you envision integrating these diagnostic models into clinical workflows, and what are the anticipated challenges?

Have authors considered to use ensemble learning for these 3 models? Will that increase the performance?

**Strengths:**

The paper successfully integrates CNNs with self-attention mechanisms, specifically Swin-Transformers and ConvNeXt, to enhance classification performance.

A two-step classification process is employed to adeptly handle the complexities associated with glomerular diseases, achieving particularly high accuracy during the initial classification stage.

**Summary Of The Paper:**

The paper focuses on the classification of 12 common types of glomerulonephritis using convolutional neural networks (CNNs) and self-attention mechanisms. The study utilizes PAS-stained sections from renal biopsies, bypassing the need for additional modalities like immunohistology or electron microscopy. A two-step classification process was implemented: first, identifying sclerosed versus non-sclerosed glomeruli, then classifying non-sclerosed glomeruli into 12 categories of glomerulonephritis. Initial classifications achieved high accuracy (97% balanced accuracy and 0.96 AUC), while subsequent classification of non-sclerosed glomeruli into specific types showed balanced accuracy up to 79.5% and an average AUC of 0.76.

**Weaknesses:**

The paper appears to be incomplete; for instance, there is a reference missing where it states: “The glomeruli in the images were automatically detected using segmentation models described in xxx, implemented by the co-authors, and reviewed by expert pathologists.”

While the initial classification results are impressive, the classification of specific types of glomerulonephritis exhibits variability and lower performance in some classes. Authors should go detailed discovery for these types.

The presentation of the model architecture lacks clarity, which could be challenging for readers without a background in deep learning.

The font size in some figures is unreadable.

---

### Official Review · Reviewer_rTk1 · 2024-08-11
**Practical paper with nice explanation of issue, methods, and how fini=dings of this paper can be use in the clinical field.**

**Overall Rating:** 8
**Confidence:** 4

**Other Quality Metrics:**

a) great
b) excellent
c) great
d) excellent

**Questions For The Authors:**

Please provide the prevalence of images in each class for both classification tasks discussed in the paper.
Could you please clarify if multiple images from the same subject were present in the data? If so, how do you handle it to prevent data leakage?

**Strengths:**

The innovation of this paper lies in its application of deep learning models, specifically ConvNeXt and Swin Transformers, to classify glomerulonephritis directly from PAS-stained sections without relying on additional diagnostic modalities such as immunostains or electron microscopy.

**Summary Of The Paper:**

The paper discusses a study on the classification of glomerulonephritis using deep learning models, specifically focusing on the ConvNeXt and Swin Transformer models. The study involves using complete microscopic images of renal biopsies, which are processed and annotated by experts to extract and label glomerular crops. The classification process is sequential, first distinguishing between sclerosed and non-sclerosed glomeruli and then further classifying the non-sclerosed glomeruli into 12 different types of glomerulonephritis. The models are evaluated using various performance metrics, including balanced accuracy. The study highlights the challenges in achieving high accuracy due to the complexity of diagnosing each pathology and the need for additional diagnostic techniques like immunofluorescence. Despite these challenges, the study establishes a baseline for this classification task and demonstrates the potential of deep learning models in nephropathology.

**Weaknesses:**

N/A

---

### Decision · Program_Chairs · 2024-09-23

Accept